

# Study on soil hydraulic properties of slope farmlands with different degrees of erosion degradation in a typical black soil region

Jianjun Mai[1,2], Zijun Wang[1,2], Feinan Hu[1,3], Jinghua Huang[1,3] and Shi-wei Zhao[1,2,3]

[1] The Research Center of Soil and Water Conservation and Ecological Environment, Chinese Academy of Sciences and Ministry of Education, Yangling, Shaanxi, China
[2] University of Chinese Academy of Sciences, Beijing, China
[3] Institute of Soil and Water Conservation, Northwest Agriculture & Forestry University, Yangling, Shaanxi, China

## ABSTRACT

In order to explore the impact of soil erosion degradation on soil hydraulic properties of slope farmland in a typical black soil region, typical black soils with three degrees of erosion degradation (light, moderate and heavy) were selected as the research objects. The saturated hydraulic conductivity, water holding capacity and water supply capacity of the soils were analyzed, as well as their correlations with soil physicochemical properties. The results showed that the saturated hydraulic conductivity of black soils in slope farmlands decreased with erosion degradation degree, which was higher in 0–10 cm soil layer than in 10–20 cm soil layer. The water holding capacity and water supplying capacity of typical black soils also decreased with the increase of erosion degradation degree, and both of them were stronger in the upper soil than in the lower soil. With the aggravation of erosion degradation of black soils, soil organic matter content decreased while soil bulk density increased, leading to the decline of soil hydraulic conductivity. The increase of soil bulk density and the decrease of contents of organic matter and >0.25 mm water stable aggregates were the main factors leading to the decrease of soil water holding capacity. These findings provide scientific basis and basic data for rational utilization of soil water, improvement of land productivity and prevention of soil erosion.

## INTRODUCTION

Soil hydraulic properties can usually be characterized by soil infiltration performance, soil water characteristic curve and soil water content, which are the basis for evaluating soil water conservation (*Huo et al., 2018*). Soil saturated hydraulic conductivity (Ks) affects surface water infiltration and runoff and sediment yield (*Fares, Aiva & Nkedi-Kizza, 2000*; *Masís-Meléndez et al., 2014*; *Wu et al., 2016*), which is an important parameter reflecting soil infiltration performance. The higher the saturated hydraulic conductivity, the better the soil infiltration performance. Increasing soil saturated hydraulic conductivity can delay surface runoff caused by precipitation, thus reducing soil erosion. Soil water characteristic

Corresponding authors
Jinghua Huang,
jhhuang@nwsuaf.edu.cn
Shi-wei Zhao, swzhao@ms.iswc.ac.cn

curve provides an important basis for evaluating soil water holding capacity and soil water availability, which reflects the relationship between soil porosity and soil water content. Therefore, all factors affecting soil pore conditions and water characteristics, such as soil texture, soil structure, soil bulk density and soil porosity, will have an impact on soil water characteristic curve (*Lei, Yang & Xie, 1988*; *Shao, Wang & Huang, 2006*; *Tang, 2017*).

The black soil region in Northeast China covers an area of 1.09 million ha. It is distributed in the temperate zone of high latitude. Due to the lush vegetation and cold winter in this region, the decomposition speed of organic matter is slow, and thus it is conductive to forming humus on the surface soil, resulting in the high organic matter content and good fertility of black soil. According to previous survey results, the content of soil organic matter is 20–40 g/kg in cultivated lands in black soil region (*Mu et al., 2020*; *Wei & Meng, 2017*). Therefore, it has become an important commodity grain production base in China, which is known as the "grain warehouse" in China (*Chinese Academy of Sciences, 2021*). It has been documented that the increase of organic matter content can increase the content of water-stable macroaggregates, improve soil structure, and then improve soil infiltration performance and soil hydraulic properties (*Wang et al., 2016*). However, in the past several decades, serious soil erosion occurred in sloping farmlands in black soil region, mainly caused by unreasonable farming measures (*He et al., 2018*; *Wei et al., 2018*; *Zhang & Liu, 2020*). It makes the black soil layer more and more thin (the average soil layer thickness has dropped from 60–80 cm in the 1950s to 20–30 cm in 2010) (*Yang, Zheng & Wang, 2016*), with the decreases of soil organic matter content (the organic matter content of surface soil decreases at an average annual rate of 5‰) (*Zhang, Liu & Zhao, 2018*), and the deterioration of soil porosity, infiltration capacity and water holding capacity (*Liu & Yan, 2009*; *Zhang et al., 2015*). Finally, these changes result in weak conductivity and low utilization efficiency of agricultural water resources (*Wei et al., 2019*). Therefore, investigating soil hydraulic properties of sloping farmland under different soil erosion degradation in black soil region can provide theoretical basis for guiding the improvement of soil water storage and conservation capacity, and enhancing the efficient use of agricultural water resources, which is of great significance for agricultural sustainable development in black soil region.

Soil hydraulic properties in black soil region of Northeast China have attracted considerable attention. For instance, it has been found that hedgerow can improve soil structure, increase soil infiltration and reduce surface runoff in black soil region of Northeast China (*Liu et al., 2017b*). *Zhai et al. (2016)* compared the accuracy of the Brooks-Corey (BC) model and the Van-Genuchten (VG) model in simulating soil water characteristic curves of black soils, and indicated that VG model was more suitable for simulating soil water characteristic curves of black soils with different erosion degrees. However, previous studies focused more on the influencing factors and improvement measures of black soil infiltration performance, as well as the model simulation of water characteristic curve and the effectiveness of soil moisture, while the understanding of water characteristic curve, water holding capacity and water supply capacity of black soils in sloping farmland with different erosion and degradation degrees are relatively limited. Based on this, our study selected typical black soils from sloping farmlands with different

erosion and degradation degrees (light, moderate and heavy erosion) in northeast China as the research objects, by determining soil saturated hydraulic conductivity, water holding capacity and water supply capacity, and analyzing their correlations with soil physicochemical properties, to clarify the influence mechanism of black soil erosion and degradation on soil hydraulic properties. We hypothesized that: (1) With the aggravation of soil erosion degradation, soil saturated hydraulic conductivity, water holding capacity and water supply capacity reduce continuously; (2) The aggravation of soil erosion degradation affects soil hydraulic properties mainly through decreasing soil organic matter content and affecting soil texture.

## MATERIALS AND METHODS

### The study area

The study region located in Keshan Experimental Station of Heilongjiang Province Hydraulic Research Institute (125°49′42″E, 48°3′33″N) in Keshan County, Qiqihar City, Heilongjiang Province, China (Fig. 1). The landform of this area is overflowing with rivers and hills, with gentle and long slopes, and hilly terrain accounts for 80% of the total area. It is influenced by cold temperate continental monsoon climate. The annual average temperature is 2.4 °C, the frost-free period is about 122 days, and the annual average precipitation is about 500 mm. More than 70% of the rainfall is concentrated between June and September, and the rain and heat are in the same period. The main soil type in this area is typical black soils, and topsoil depth is about 20 cm. The cropping system is one crop a year, soybean and corn rotation.

### Selection of sampling plots

Slope farmlands in a back soil region have suffered from soil erosion, which leads to thinning of black soil layer, decrease of soil nutrients and crop yield (He & Xiao, 2022; Liu & Yan, 2009). It has been reported that soil erosion intensity of slope farmlands in black soil region can be categorized according to slope degree (Han & Guo, 2017; Yang, Wang & Xie, 2009). In our study, we further calculated soil loss speed and erosion modulus based on slope degree (Kang, Liu & Liu, 2017; Yan & Tang, 2005), and also investigated black soil layer thickness and crop yield (Wang, Liu & Wang, 2009; Zhang & Liu, 2020), to define soil erosion degree of slope farmlands in the black soil region. Finally, we selected three sampling sites with different degrees of erosion degradation (light, moderate and heavy erosion), based on the comprehensive consideration of slope degree, black soil layer thickness, crop yield, soil loss speed and erosion modulus. The detailed information of the three sampling sites can be seen in Table 1, and the location of these sites can be seen in Fig. 1.

### Soil sampling

Field experiments were approved by the Heilongjiang Province Hydraulic Research Institute (12230000414003295L) and after we obtained oral permission from the administrator (Mr. Xujun Liu, the head of Keshan Experimental Station of Heilongjiang Province Hydraulic Research Institute), we collected the soil samples in June of 2022. Soil

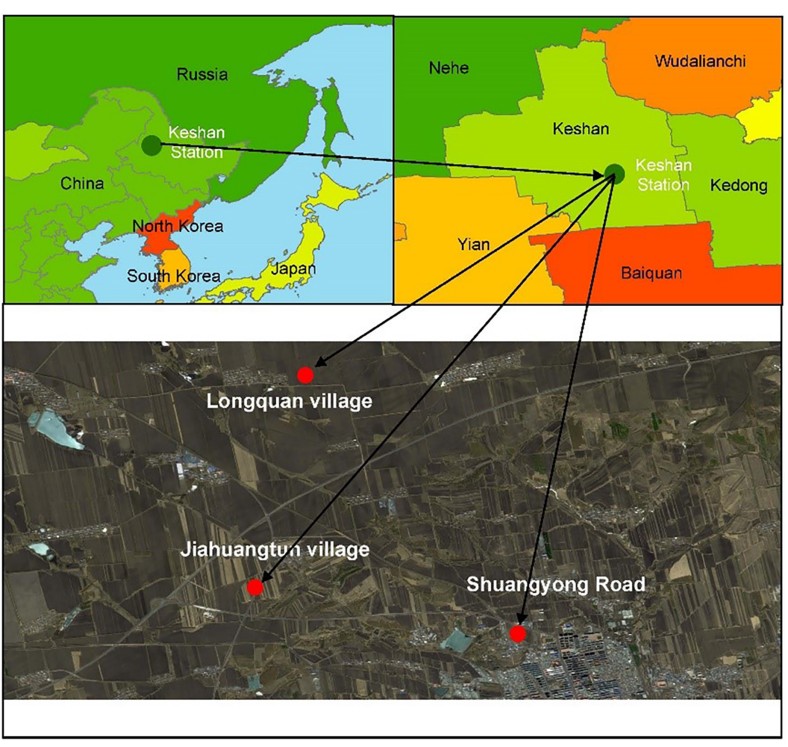

**Figure 1 Location of sampling sites.** Map source: https://map.qq.com/.

**Table 1 Basic information of sampling sites.**

| Erosion degradation degree | Plot location | Slope degree/(°) | Soil depth/cm | Corn yield/ (kg•ha⁻¹) | Soil loss speed/ (mm•a⁻¹) | Erosion modulus/ (t•km⁻²•a⁻¹) |
|---|---|---|---|---|---|---|
| L | Longquan village | 3.4 | 43 | 12,000 | 2.51 | 3,003 |
| M | Jiahuangtun village | 6.2 | 27 | 10,500 | 4.58 | 5,480 |
| H | Shuangyong road | 10.6 | 18 | 8,250 | 7.83 | 9,368 |

**Note:**
L, lightly eroded soils; M, moderately eroded soils; H, heavily eroded soils.

samples were collected from the lightly, moderately and seriously eroded plots. Three sampling quadrats were randomly selected from each sample plot. In each quadrat, soil samples were collected from 0–10 and 10–20 cm soil layers, respectively, by plum blossom five-point sampling method. Undisturbed soil samples and cutting ring soil samples were also collected from the two soil layers.

## Soil properties determination

Soil bulk density was measured by the cutting ring method (*Blake & Hartge, 1986*). The mechanical composition of soil was measured by the pipette method (*Day, 1965*). Soil water-stable aggregates were determined by the wet sieve method (*ISSAS, 1978*). Soil organic matter content was determined by potassium dichromate external heating method

(*Kononova, 1961*). Soil total nitrogen content was determined by the semi-micro Kjeldahl method (*Bremner, 1960*). Soil available phosphorus was extracted by 0.5 mol/L sodium bicarbonate solution and the concentration in extracts was determined by molybdenum antimony colorimetry method (*Olsen et al., 1954*). Soil available potassium was extracted by ammonium acetate and the concentration in extracts was determined by flame spectrophotometry method (*Pansu & Gautheyrou, 2007*). The saturated hydraulic conductivity of soil was measured by constant head method (*Klute & Dirksen, 1986*). The characteristic curve of soil moisture was measured by centrifuge method (*Soil Moisture Determination Method, 1986*).

## Fitting model

Due to the wide range of soil texture and high fitting degree of the linear type with measured data (*Van Genuchten, 1980*), the Van Genuchten (VG) model has been widely used for estimating soil water characteristic curve, especially in black soil region (*Gao, Gu & Li, 2018*; *Wang et al., 2018*). Therefore, in this study, the VG model was adopted, and its expression formula (*Lei, Yang & Xie, 1988*) is as follows:

$$\theta = \theta_r + \frac{\theta_s - \theta_r}{\left(1 + |\alpha \cdot h|^n\right)^m}$$

In the above formula, $\theta$ is the volume moisture content of soil under suction(h); $\theta_r$ is the permanent wilting point; $\theta_s$ is the saturated volume moisture content; $\alpha$ is the suction value related to the inlet air value, which is equal to the reciprocal of the inlet air value, and the inlet air value of soil is related to the soil texture. Generally, the inlet air value of heavy clay soil is larger, while that of light soil or well-structured soil is smaller; $h$ is soil water suction; $n$ and $m$ are curve shape parameters, $n$ reflects the change of soil moisture content with soil water suction, and the value of $n$ determines the slope of soil water characteristic curve. The larger the value of $n$, the slower the slope of the curve, taking $m = 1 - \frac{1}{n}$.

The formula of specific water capacity is:

$$C(h) = \frac{(\theta_s - \theta_r)mn\alpha(\alpha h)^{n-1}}{\left(1 + (\alpha h)^n\right)^{m+1}}$$

## Data processing and analysis

Soil water characteristic curve was fitted by RETC software (https://www.pc-progress.com/en/default.aspx?retc). The differences in soil properties (*e.g.*, soil bulk density, soil mechanical composition, soil organic matter content and saturated hydraulic conductivity) among sloped farmlands with different degree of erosion and degradation were analyzed by one-way ANOVA analysis, and the correlations between soil physicochemical properties (soil organic matter content, soil bulk density, sand, silt, clay) and water characteristic parameters ($\alpha$, n, C (100), Ks) were analyzed by Pearson correlation analysis, using SPSS 17.0 software (SPSS Inc, Chicago, IL, USA).

## RESULTS

### Saturated hydraulic conductivity of black soils

Soil saturated hydraulic conductivity is an important parameter reflecting soil infiltration performance. The greater the infiltration performance of soils, the greater its water retention potential. As shown in Fig. 2, the saturated hydraulic conductivity of lightly eroded (L) slope farmland soils was between 0.04–0.11 mm/min, which was higher than those of moderately eroded (M) (0.02–0.05 mm/min) and heavily eroded (H) slope farmland soils (0.01–0.04 mm/min), with a decrease range of 63.6–75%. The saturated hydraulic conductivity of soil decreased with the increase of depth, 0.04–0.11 mm/min in 0–10 cm soil and 0.01–0.05 mm/min in 10–20 cm soil, with a decrease range of 54.5–75%. The saturated hydraulic conductivity of lightly, moderately and heavily eroded slope farmland soils decreased by 63.6%, 60% and 75%, respectively, with the increase of depth. With the aggravation of soil erosion and degradation, soil permeability and hydraulic conductivity decreased.

### Water holding capacity and water supply capacity of black soils

The centrifuge method was used to measure the water content of black soils in slope farmlands with different degrees of erosion degradation after natural water absorption saturation and soil water balance under different rotating speed (suction value). Then, VG equation was used to fit it. The parameter values are shown in Table 2. The correlation coefficient $R^2$ was above 0.7594. The VG equation can well simulate the water characteristic curves of black soils with different degradation degrees, as shown in Fig. 3.

The difference between saturated water content $\theta_s$ and permanent wilting point $\theta_r$ can characterize the water holding capacity of soil. The greater the difference, the stronger the water-holding capacity of the soil. The differences of saturated water content and permanent wilting point of 0–10 and 10–20 cm soil layers were 0.4418 and 0.4245 respectively in lightly eroded sampling plot (L), 0.4076 and 0.3880 respectively in moderately eroded sampling plot (M), and 0.3783 and 0.3662 respectively in heavily eroded sampling plot (H). It can be seen that the water holding capacity of lightly eroded farmland soil was the strongest, followed by moderately eroded farmland soil, and the water holding capacity of the upper soil was stronger than that of the lower layer. Therefore, with the aggravation of erosion degradation, the water holding capacity of black soils decreased.

Table 2 and Fig. 3 indicated that the parameter $n$ characterizing the shape of water characteristic curve gradually decreased with the aggravation of black soil erosion and degradation, and the slope of water characteristic curve of heavily eroded farmland black soils was the steepest, followed by moderately eroded soil and finally lightly eroded soil. The $\alpha$ value of black soils listed as lightly eroded farmland < moderately eroded farmland < heavily eroded farmland. It can also be seen that with the aggravation of erosion, the content of soil clay gradually decreased, and the content of soil sand increased, which reduces the water-holding capacity of soil (Table 3).

The results showed that under the same soil water suction, the specific water capacity of 0–10 cm soil layer was larger than that of 10–20 cm soil layer, and the specific water

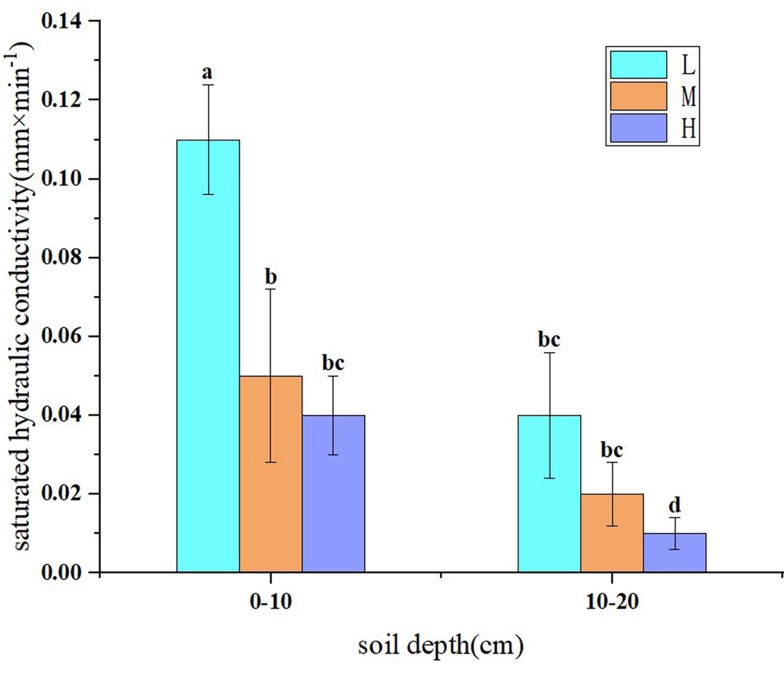

**Figure 2 Saturated hydraulic conductivity of soils in slope farmlands with different erosion and degradation degrees.** Note: L, lightly eroded soils; M, moderately eroded soils; H, heavily eroded soils. Values are means of three replicates ± SD and different lowercase letters indicate significant differences (Duncan's multiple range test, $p < 0.05$).

**Table 2 Fitting parameters of VG model of water characteristic curve.**

| Erosion degradation degree | Soil depth/cm | $\theta_r$/(cm³•cm⁻³) | $\theta_s$/(cm³•cm⁻³) | $\alpha$/(cm⁻¹) | n | $R^2$ |
|---|---|---|---|---|---|---|
| L | 0–10 | 0.0966 ± 0.0010a | 0.5384 ± 0.0029a | 0.0122 ± 0.0004c | 1.4356 ± 0.0087ab | 0.9939 |
| | 10–20 | 0.0947 ± 0.0012ab | 0.5192 ± 0.0030b | 0.0117 ± 0.0007c | 1.445 ± 0.0147a | 0.9931 |
| M | 0–10 | 0.0927 ± 0.0005bc | 0.5004 ± 0.0011c | 0.0129 ± 0.0001abc | 1.428 ± 0.0030abc | 0.9829 |
| | 10–20 | 0.091 ± 0.0002c | 0.479 ± 0.0002d | 0.0127 ± 0.0001bc | 1.4212 ± 0.0013abc | 0.8226 |
| H | 0–10 | 0.0862 ± 0.0013d | 0.4645 ± 0.0023e | 0.0139 ± 0.0006ab | 1.4134 ± 0.0095bc | 0.7594 |
| | 10–20 | 0.0842 ± 0.0016d | 0.4504 ± 0.0032f | 0.0141 ± 0.0010a | 1.4045 ± 0.0181c | 0.7053 |

**Note:**
L, lightly eroded soils; M, moderately eroded soils; H, heavily eroded soils. Values are means of three replicates ± SD and different lowercase letters indicate significant differences (Duncan's multiple range test, $p < 0.05$).

capacity of the same soil layer list as L > M > H (Fig. 4). The specific water capacity of 0–10 and 10–20 cm soil layer in M were 7.52% and 10% lower than those in L, and the specific water capacity of 0–10 and 10–20 cm soil layer in H were 7.75% and 5.73% lower than those in M, respectively (Fig. 4). Therefore, soil erosion and degradation reduce the water supply capacity of soil.

## Correlations between soil physicochemical properties and water characteristic parameters

The relationship between soil physicochemical properties and soil erodibility was not specifically analyzed in this study. However, our results showed that with the

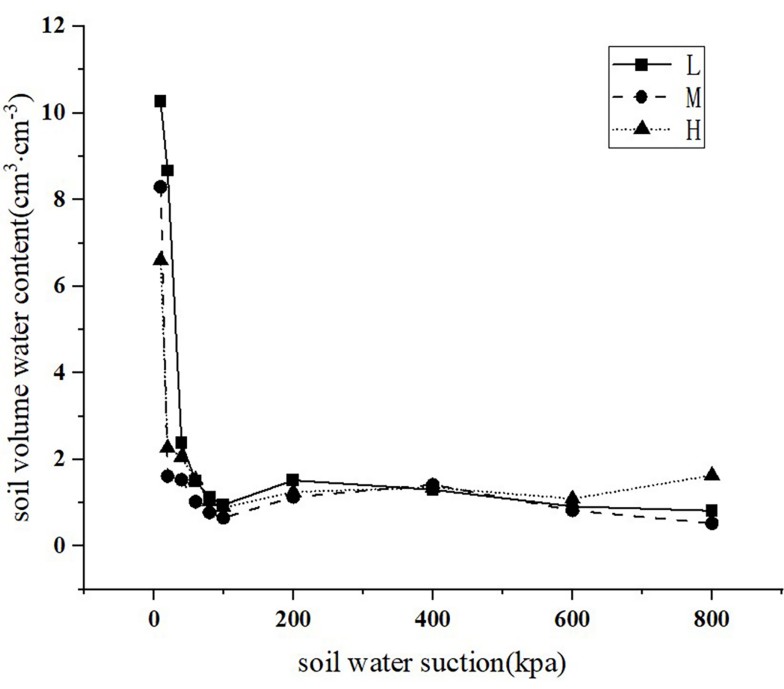

**Figure 3 Characteristic curves of soil moisture in slope farmlands with different erosion and degradation degrees.** Note: L, lightly eroded soils; M, moderately eroded soils; H, heavily eroded soils.

**Table 3 Physicochemical properties of black soil in slope farmlands with different erosion and degradation degrees.**

| Erosion degradation degree | Soil depth/ cm | Organic matter content/(g·kg⁻¹) | Soil bulk density/ (g·cm⁻³) | Sand/% | Silt/% | Clay/% | >0.25 mm water-stable aggregates/% | TN/(g·kg⁻¹) | OP/(mg·kg⁻¹) | AK/(mg·kg⁻¹) |
|---|---|---|---|---|---|---|---|---|---|---|
| L | 0–10 | 3.71 ± 0.78a | 1.09 ± 0.1c | 22.14 ± 1.07c | 40.82 ± 0.23a | 37.03 ± 0.94a | 80.96 ± 1.68b | 1.019 ± 0.08a | 52.303 ± 7.23a | 273.765 ± 5.23a |
|   | 10–20 | 3.21 ± 0.08ab | 1.15 ± 0.04bc | 22.46 ± 0.3c | 40.87 ± 1.33a | 36.68 ± 1.26ab | 84.86 ± 0.82a | 0.719 ± 0.09c | 47.852 ± 2.27ab | 255.040 ± 5.38b |
| M | 0–10 | 3.35 ± 0.17ab | 1.21 ± 0.12abc | 28.41 ± 0.53b | 34.57 ± 0.24b | 37.02 ± 0.42a | 78.69 ± 1.58b | 0.916 ± 0.01a | 40.847 ± 1.67bc | 248.909 ± 5.27b |
|   | 10–20 | 3.18 ± 0.28ab | 1.29 ± 0.04ab | 27.95 ± 0.51b | 34.57 ± 0.46b | 37.48 ± 0.08a | 80.50 ± 0.27b | 0.836 ± 0.03b | 35.698 ± 2.53cd | 231.615 ± 1.46c |
| H | 0–10 | 2.67 ± 0.11b | 1.32 ± 0.02a | 36.54 ± 1.77a | 28.82 ± 1.4c | 34.64 ± 1.11bc | 73.54 ± 0.63c | 0.807 ± 0.02bc | 29.264 ± 3.05de | 226.999 ± 9.18cd |
|   | 10–20 | 2.53 ± 0.05c | 1.37 ± 0.04a | 37.24 ± 1.46a | 28.47 ± 2.13c | 34.28 ± 1.53c | 75.63 ± 2.21c | 0.703 ± 0.01c | 21.524 ± 1.79e | 218.716 ± 4.53d |

**Note:**
Sand (2–0.02 mm), silt (0.02–0.002 mm) and clay (<0.002 mm).

intensification of soil erosion and degradation of slope farmlands, the contents of soil organic matter, >0.25 mm water-stable aggregates, silt and clay decreased, while soil bulk density and sand content increased, as shown in Table 3. These results indicated the close relations between soil erodibility and soil physicochemical properties for slope farmlands in black soil region.

Soil hydraulic properties are further affected by soil physicochemical properties. The correlations between the physicochemical properties and water characteristic parameters of surface soil in slope farmlands with different erosion and degradation degrees were analyzed (Table 4).

Parameter $\alpha$ was significantly negatively correlated with soil organic matter and clay content ($P < 0.05$), which was extremely significantly negatively correlated with >0.25 mm

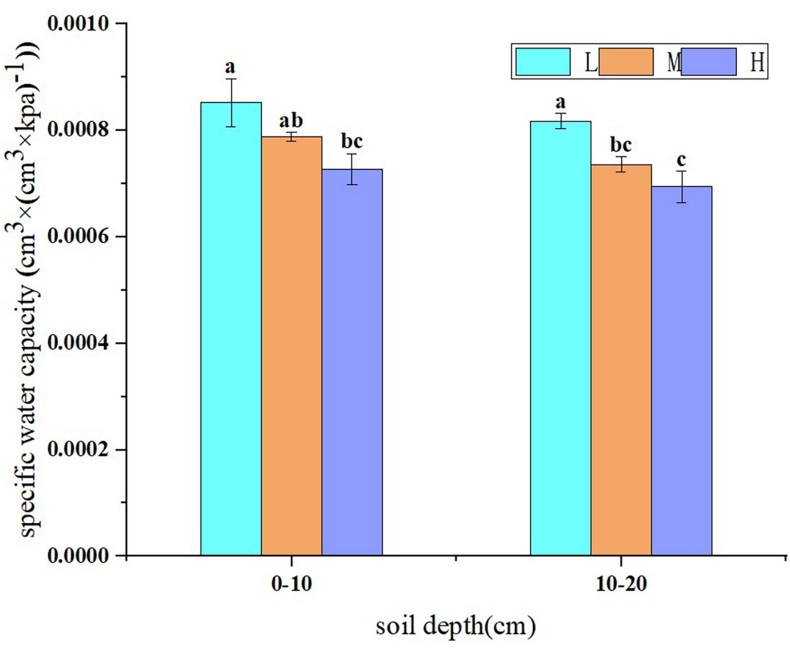

**Figure 4 Specific water capacity of soil in slope farmlands with different erosion and degradation degrees.** Note: L, lightly eroded soils; M, moderately eroded soils; H, heavily eroded soils. Values are means of three replicates ± SD and different lowercase letters indicate significant differences (Duncan's multiple range test, $p < 0.05$).

**Table 4 Pearson correlation coefficient between soil physicochemical properties and water characteristic parameters.**

| Water characteristic parameter | Physicochemical properties of soil | | | | | |
|---|---|---|---|---|---|---|
| | Organic matter content | Soil bulk density | Sand | Silt | Clay | >0.25 mm water-stable aggregates |
| α | −0.820* | 0.874* | 0.981** | −0.976** | −0.822* | −0.962** |
| n | 0.797 | −0.920** | −0.943** | 0.954** | 0.727 | 0.887* |
| C(100) | 0.904* | −0.997** | −0.916* | 0.931** | 0.689 | 0.731 |
| Ks | 0.786 | −0.834* | −0.621 | 0.645 | 0.411 | 0.292 |

Notes:
* A significant correlation at level 0.05 (bilateral).
** A very significant correlation at level 0.01 (bilateral).
Values are means of three replicates ± SD and different lowercase letters indicate significant differences (Duncan's multiple range test, $p < 0.05$).

water-stable aggregates and silt content ($P < 0.01$), while it was significantly positively correlated with bulk density ($P < 0.05$), and extremely significantly positively correlated with sand content ($P < 0.01$) (Table 4). Parameter $n$ was negatively correlated with soil bulk density and sand content ($P < 0.01$), and positively correlated with >0.25 mm water-stable aggregates and silt content ($P < 0.01$), but it was not correlated with organic matter and clay content (Table 4). The correlation between specific water capacity and soil bulk density and silt content were very significant. The specific water capacity of soil decreased with the increase of soil bulk density and the decrease of silt content (Table 4). In addition, soil specific water capacity was significantly positively correlated with soil organic matter while negatively correlated with sand content ($P < 0.05$). There was a significant negative correlation between saturated hydraulic conductivity and soil bulk density ($P < 0.05$), but

no significant correlation was found between saturated hydraulic conductivity and soil organic matter, sand, silt, clay and >0.25 mm water-stable aggregates content.

## DISCUSSION

Soil erodibility is closely related to soil physicochemical properties (*Jiang, Pan & Yang, 2004*; *Yang, Yang & Ma, 2014*). A large number of studies have shown that soil erodibility is negatively correlated with the contents of organic matter and >0.25 mm water-stable aggregates in soil, and positively correlated with soil bulk density (*Fan, Zhu & Shangguan, 2023*; *Lu, 2022*; *Lv, 2021*; *Wang, Cui & Zhao, 2017*). Soil physicochemical properties, such as soil organic matter content, mechanical composition, bulk density, pore distribution, greatly changed with the intensification of soil erosion and degradation, which could have significant effects on soil saturated hydraulic conductivity and soil erodibility. With the aggravation of erosion degree, soil sand content increased, clay content and organic matter content decreased, soil aggregate particles were broken, and aggregate stability decreased (*Ai, 2013*; *Gao, Gu & Li, 2018*). The saturated hydraulic conductivity of soil increased with the increase of soil organic matter and total porosity, but decreased with the increase of soil bulk density (*Mao, Huang & Shao, 2019*; *Wang et al., 2016*; *Zhang, Zhao & Hua, 2009*). Consistent with our first hypothesis, with the aggravation of black soils erosion and degradation, the saturated hydraulic conductivity of soil decreased, because soil organic matter and >0.25 mm water-stable aggregates content gradually decreased with soil erosion and degradation, which increased soil bulk density and led to the decrease of soil water permeability. Our results were consistent with the findings by *Zhang et al. (2015)* and *Jing, Liu & Ren (2008)*. In addition, previous studies have shown that the destruction of soil structure will lead to the decrease of soil infiltration rate and present a significant positive correlation (*Yang, Zhao & Lei, 2006*; *Yang, Wu & Zhao, 2009*). We found that the saturated hydraulic conductivity of lightly eroded topsoil was significantly higher than that of moderately eroded topsoil, while there was no significant difference between the saturated hydraulic conductivity of moderately eroded topsoil and that of heavily eroded topsoil. This may be due to the significant destruction of soil structure from light to moderate erosion, resulting in a significant decrease in soil infiltration performance to a very low level. From moderate to heavy erosion, the damage degree of soil structure is reduced, so that the soil infiltration performance is not significantly reduced.

Previous results have shown that compared with other models, the VG model has the highest accuracy for simulating soil water characteristic curve (*Deng et al., 2016*; *Wang et al., 2018*; *Zhang et al., 2022*). In this study, the VG model was used to simulate the water characteristic curve of black soils, and the fitting correlation coefficients ($R^2$) were all between 0.7594 and 0.9939. Therefore, this model can be effective in fitting the relationship between water content and water suction of black soils in slope farmlands with different erosion and degradation degrees. Compared with the VG model of lightly and moderately eroded soil, the $R^2$ value of the heavily eroded soil VG model was much lower. The reason may be that the sand content of heavily eroded soil is significantly higher than that of lightly and moderately eroded soil, so that the water holding capacity of heavily eroded soil is lower. The soil moisture content decreased significantly with the increase of water

suction, resulting in a small change of soil moisture content with water suction in the middle and late centrifugation period. Therefore, the VG model $R^2$ value of heavily eroded soil with higher sediment content is smaller.

With the aggravation of soil erosion and degradation degree, the difference between soil saturated water content $\theta_s$ and permanent wilting point $\theta_r$ decreased, as well as shape parameter $n$, indicating that soil water holding capacity was weakened. That might be because with the aggravation of soil erosion, the contents of soil organic matter and clay decrease and the content of sand increases, which eventually leads to the decrease of soil water holding capacity (Zhai et al., 2016). Ma, Fu & Luo (2017) indicated that the difference between soil saturated water content $\theta_s$ and permanent wilting point $\theta_r$ could characterize the water holding capacity of soil, with greater difference reflecting stronger water holding capacity of the soil. Dong et al. (2017) found that the larger the fitting parameter $n$ of the VG model, the better the soil water retention capacity. Therefore, the water holding capacity of typical black soils decreases with the aggravation of black soils erosion and degradation.

It has been found that the parameters $\alpha$ and $n$ of VG model water characteristic curve can reflect the water holding capacity of soil, and the smaller the $\alpha$ value and the larger the $n$ value, the better the water holding capacity of soil (Ma, Fu & Luo (2017); Pan, Lei & Zhang, 2007; Wang et al., 2018). Soil water holding capacity is mainly affected by soil basic physicochemical properties such as soil bulk density, organic matter content, soil texture, soil porosity and so on. Soil water holding capacity positively correlated with soil texture and porosity, and negatively correlated with soil bulk density (Liu et al., 2017a; Zhao, Zhou & Wu, 2002). The results of this study showed that the parameter $\alpha$ was negatively correlated with the contents of organic matter, silt, clay and >0.25 mm water-stable aggregates in soil ($P < 0.05$), but positively correlated with soil bulk density and sand content ($P < 0.05$). Parameter $n$ was negatively correlated with soil bulk density and sand content ($P < 0.01$), and positively correlated with silt and >0.25 mm water-stable aggregates content ($P < 0.01$), but did not correlate with soil organic matter and clay contents. Our results provided evidence that soil erosion and degradation led to the decreases of the contents of soil organic matter and >0.25 mm water-stable aggregates, while resulted in the increase of soil bulk density, which consequently decreased soil water holding capacity. Therefore, soil bulk density and the contents of organic matter and >0.25 mm water-stable aggregates were the main factors affecting soil water holding capacity. In Table 3, in H erosion degradation class silt content was lower than L&M in two depths of samples. The reason may be that soil erosion will lead to the fragmentation of soil aggregate particles, and the small particles generated after the fragmentation of micro-aggregates are carried away by rain and wind, resulting in the imbalance of soil aggregates. The decrease of aggregate stability in turn resulted in the intensification of surface runoff and soil erosion, the decrease of soil particle content and coarser texture (Ai, 2013). Earlier studies have also showed that the clay and silt contents of black soils decreased with the increase of erosion degree (Zhai et al., 2016; Gao, Gu & Li, 2018), which supported our results.

The specific water capacity when the soil water suction is 100 kPa (C (100)) can well measure the water supply capacity of soil (*Liu et al., 2019*). There is research indicated that specific water capacity is a useful index to measure the amount of water that can be released by soil to supply plant absorption (*Liu et al., 2017a*). The greater the specific water capacity, the stronger the soil water supply capacity and drought resistance. In this study, the specific water capacity of soil decreased with soil erosion and degradation, which indicated that soil erosion and degradation reduced the water supply capacity of soil, mainly due to the fact that soil with low degree of erosion and degradation has higher organic matter content, better soil structure and higher water absorption capacity, thus making the water supply capacity stronger (*Ma et al., 2005*).

Finally, it should be noted that the accuracy of the VG model varies with changes in soil texture and physicochemical properties. For example, when the sand content in the soil is high (more than 50%), the VG model has poor fitting effect (*Zhan, Li & Yu, 2022*). Therefore, although our results have demonstrated that the VG model can be effective in simulating soil water characteristic curves of black soils in slope farmlands with different erosion and degradation degrees, caution is needed in applying and extending the conclusions drawn from this model.

## CONCLUSIONS

The water characteristics of black soils in sloping farmlands with different degrees of erosion degradation have seldom reported in the past. Our study investigated the saturated hydraulic conductivity, water holding capacity and water supply capacity of black soils in lightly, moderately and seriously eroded slope farmlands, fitted them by the VG model, and explored their correlations with soil physicochemical properties. The results support our hypotheses that the aggravation of erosion and degradation of black soil in slope farmlands coarses soil texture, reduces the contents of organic matter and >0.25 mm water-stable aggregates, and increases soil bulk density, which leads to the decrease of soil saturated hydraulic conductivity and weakens soil water holding capacity and water supply capacity. These findings provide scientific basis and basic data for rational utilization of soil water, improvement of land productivity and prevention of soil erosion. Therefore, improving soil water characteristics of sloping farmland in black soil region can not only increase soil infiltration, reduce surface runoff and erosion, enhance water storage and moisture conservation capacity, but also provide theoretical basis for efficient use of agricultural water resources, which is of great significance for agricultural sustainable development in black.

### Funding

This work was supported by the Joint Funds of the National Natural Science Foundation of China (No. U2243225). The funders had no role in study design, data collection and analysis, decision to publish, or preparation of the manuscript.

### Grant Disclosures

The following grant information was disclosed by the authors:
National Natural Science Foundation of China: U2243225.

### Competing Interests

The authors declare that they have no competing interests.

### Author Contributions

- Jianjun Mai conceived and designed the experiments, performed the experiments, analyzed the data, prepared figures and/or tables, authored or reviewed drafts of the article, and approved the final draft.
- Zijun Wang performed the experiments, analyzed the data, authored or reviewed drafts of the article, and approved the final draft.
- Feinan Hu analyzed the data, authored or reviewed drafts of the article, and approved the final draft.
- Jinghua Huang performed the experiments, authored or reviewed drafts of the article, and approved the final draft.
- Shi-wei Zhao conceived and designed the experiments, authored or reviewed drafts of the article, and approved the final draft.

### Field Study Permissions

The following information was supplied relating to field study approvals (*i.e.*, approving body and any reference numbers):

Field experiments were approved by Heilongjiang Province Hydraulic Research Institute (12230000414003295L).

### Data Availability

The raw data is available in the Supplemental File.

### Supplemental Information

Supplemental information for this article can be found online at http://dx.doi.org/10.7717/peerj.15930#supplemental-information.

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
