# Peer review of "Study on soil hydraulic properties of slope farmlands with different degrees of erosion degradation in a typical black soil region"

_PeerJ, doi:10.7717/peerj.15930_

## Round 0.1 · original submission · Major Revisions

Dear Dr Mai,

We received two evaluations of your manuscript. Both reviewers agree on the interest and value of your study but suggest relevant modifications. From my side, I would recommend some more descriptions of the black soil for readers not familiar with this soil type. For instance, I would be curious to know what is soil organic matter in the soil profile, to which extent is it affected by soil erosion and if this is possibly linked with soil hydrological characteristics.

I expect the revised version of your manuscript and the answers to the reviewers' comments.

Sincerely

Leonardo Montagnani

Reviewer 1 ·

Basic reporting

Overall manuscript is nicely compiled, and concept is appealing however, some important improvements are suggested:
• Tittle is not aligning with the present research. In place of soil water characteristics, “soil hydraulic properties” may sound more relevant.
• Abstract is fine.
• Introduction if adequate: covered background and establishing hypothetical connect.
• References are relevant and properly written.
• Check the relevance of refence of “Mollinedo et al., 2015”. It may be as below.
• M.T. Van Genuchten, A closed-form equation for predicting the hydraulic conductivity of unsaturated soils, Soil Soc. Am. J. 44 (1980) 892–898.

Experimental design

Methodology is adequately written; however, some aspects are required to address for better understanding.
• On what basis the categories of low, moderate, and high erosion are defined? Is there any parameter like carbon content/soil thickness?
• How was the degree of erodibility or erosion degradation and slope defined?
• Better to use metric units like “kg/ha” rather than “kg/hm2”, as such units are globally used.
• Sentence “In each quadrat, soil samples were collected by plum blossom five-point sampling method at the depth of 0-10 cm and 10-20 cm soil layers, and undisturbed soil samples, mixed soil samples and cutting ring soil samples were taken respectively” is unclear. Need to rewrite [line 105-107].
• In the section-“Soil Properties Determination”, methodology for N,P,K, Water stable aggregate has been explained but such data is not given anywhere in the manuscript.

Validity of the findings

• Graphical and tabular representation of findings is good.
• Findings of the present investigation have been written adequately and explained nicely referring the tables and figure.
• Major findings are on soil hydrological properties only. Inclusion of soil physical and chemical properties data may enhance the relevance and reader base of manuscript.
• Discussion part is adequate and nicely supporting the results sections.
• However, conclusion part may be improved to make more appealing. It sounds more like summary or highlight of results rather than conclusion for present investigation.

Reviewer 2 ·

Basic reporting

No comment

Experimental design

Samples were collected for three different degrees of erosion degradation but without addressing the topsoil loss per year along with the slope of each class which is essential to understand the intensity of erosion.

Validity of the findings

1. If data on total available water content (TAW), rapidly available water content (RAW), slow available water content (SAW) and unavailable water content (UAW) in the different soil layers of different degrees of erosion classes is available that data need to be presented.
2. The soil total porosity parameter in the different soil layers at the different degrees of erosion needs to be included
3. Aggregate stability needs to be presented in the results to understand the erosion of soil.
4. Replace the term residual water content with Permanent wilting point otherwise international readers will confuse.

Additional comments

4. Replace the term residual water content with Permanent wilting point otherwise it will the international readers.
5. In table-2, in H erosion degradation class silt content was lower than L& M in two depths of samples. Are there any specific reasons?
6. In line no 181 correct the α value for different degrees of erosion degradation
7. Saturated hydraulic conductivity between topsoil of L& M is huge
8. VG model R2 values were lower in the H erosion degradation degree. Reasons for such lower R2 values were not discussed.

---

## Round 0.2 · Minor Revisions

Dear Dr. Mai,

We received two evaluations of your article. While one of the reviewers was fully satisfied with the modifications done, the second is still asking for a few modifications.

I am also satisfied with the better description of black soil. I am therefore waiting for the answer to the review which is asking for additional changes.

Sincerely,

Leonardo Montagnani

Reviewer 1 ·

Basic reporting

Authors have incorporated suggestions. However, It can be accepted after the inclusion of following:

In methodology section (soil properties determination), reference should be given from methodology followed.

In the result section (correlation) authors should highlight the correlation between soil properties and erodibility as the soil organic carbon, BD, aggregation, aggregate stability etc.

Experimental design

NA

Validity of the findings

NA

Reviewer 2 ·

Basic reporting

The basic reporting of the manuscript was executed well with a logical flow of information, providing relevant background information, and clearly stating the research objectives. The authors described the study design, data collection procedures, and statistical analysis methods used.The results are presented in a clear and organised manner, utilising appropriate tables and figures. The authors have compared their findings to existing literature and provided plausible explanations.

Experimental design

Manusript falls within the aims and scope of the journal.
The experimental design is well structured, allowing for a comprehensive investigation of black soil's hydraulic properties.
The selection of site and characterization of degrees of erosion described, ensuring the relevance and representativeness of the findings

Validity of the findings

The findings of the study demonstrate the accuracy and reliability of the results.

Additional comments

The authors have failed to address the potential limitations of the VG model used in the study

---

## Round 0.3 · accepted · Accept

Dear Dr Mai,

I am pleased to inform you that I consider your paper acceptable now.

Sincerely,

Leonardo Montagnani